# Effect of Scapular Fixation on Lateral Movement and Scapular Rotation during Glenohumeral Lateral Distraction Mobilisation

**DOI:** 10.3390/healthcare11131829

**Published:** 2023-06-22

**Authors:** Carlos López-de-Celis, Santos Caudevilla-Polo, Albert Pérez-Bellmunt, Vanessa González-Rueda, Elena Bueno-Gracia, Elena Estébanez-de-Miguel

**Affiliations:** 1Faculty of Medicine and Health Sciences, Universitat International de Catalunya, 08195 Barcelona, Spain; 2ACTIUM Functional Anatomy Group, 08195 Barcelona, Spain; carlesldc@uic.es (C.L.-d.-C.); aperez@uic.cat (A.P.-B.); vgonzalez@uic.es (V.G.-R.); 3Fundació Institut Universitari per a la Recerca a l’Atenció Primària de Salut Jordi Gol i Gurina (IDIAPJGol), 08007 Barcelona, Spain; 4Faculty of Health Sciences, University of Zaragoza, 50009 Zaragoza, Spain; ebueno@unizar.es (E.B.-G.); elesteba@unizar.es (E.E.-d.-M.)

**Keywords:** glenohumeral joint, manual therapy, mobilisation, ultrasound

## Abstract

Background: Glenohumeral lateral distraction mobilisation (GLDM) is used in patients with shoulder mobility dysfunction. No one has examined the effect of scapular fixation during GLDM. The aim was to measure and compare the lateral movement of the humeral head and the rotational movement of the scapula when three different magnitudes of forces were applied during GLDM, with and without scapular fixation. Methods: Seventeen volunteers were recruited (*n* = 25 shoulders). Three magnitudes of GLDM force (low, medium, and high) were applied under fixation and non-fixation scapular conditions in the open-packed position. Lateral movement of the humeral head was assessed with ultrasound, and a universal goniometer assessed scapular rotation. Results: The most significant increase in the distance between the coracoid and the humeral head occurred in the scapular fixation condition at all three high-force magnitudes (3.72 mm; *p* < 0.001). More significant scapular rotation was observed in the non-scapular fixation condition (12.71°). A difference in scapula rotation (10.1°) was observed between scapular fixation and non-scapular fixation during high-force application. Conclusions: Scapular fixation resulted in more significant lateral movement of the humeral head than in the non-scapular fixation condition during three intensities of GLDM forces. The scapular position did not change during GLDM with the scapular fixation condition.

## 1. Introduction

In primary care, shoulder pain has an annual incidence of 17.3 per 1000 in individuals aged 45–64 years and 12.8 per 1000 in those aged 65–74 [1]. A total of 67% of the general population will experience shoulder pain in their lifetime [2], often associated with rotator cuff injuries [3]. Mobility dysfunctions in the shoulder, such as adhesive capsulitis, are estimated to be prevalent in 2% to 5% of the general population [4].

Clinical practice guidelines recommend treatments aimed at mobilising the glenohumeral joint in patients with adhesive capsulitis. The aim is to relieve pain, increase shoulder mobility, and improve physical function in these patients [5]. One of mobilisation approaches used in orthopaedic manual therapy is translatory mobilisation. This approach is widely accepted for the treatment of shoulder pathologies [6,7,8,9,10]. The movement sought is a linear translatory movement relative to a joint plane represented by the concave surface of the joint, in this case the glenoid [11]. Lateral glenohumeral distraction mobilisation (GLDM) has been found to effectively decrease pain and increase mobility in many shoulder dysfunctions [12,13,14,15,16,17,18]. Therefore, this technique is used in multiple pathologies with glenohumeral dysfunction for this purpose [19,20,21]. In clinical practice, the GLDM technique is applied with scapular fixation [17]. The aim is to have a better control of the translational motion and to prevent a scapular motion produced by the applied force [22]. It also prevents the applied force from being directed to other structures in the scapulothoracic and cervical region, which could cause harm in these regions [23,24,25]. Scapular fixation is crucial to produce a more significant distraction effect on the joint, increasing the space between the glenoid and the humeral head. A similar technique has been studied as the glenohumeral axial distraction technique, which produces a caudal slip by increasing the subacromial space [26,27,28]. A recent study demonstrated that fixation condition affected glenohumeral axial distraction mobilisation [22]. However, we do not know whether the GLDM technique produces a lateral displacement to the glenoid and its magnitude when applied clinically. We are also unaware of the differences between scapular and non-scapular fixation conditions with different mobilisation degrees, as Kaltenborn described [22].

According to Kaltenborn, the classification system for tensile force is divided into three grades. Grade I (low force) overrides the natural compression of a joint. Grade II (medium force) tightens the periarticular tissue until maximum resistance to movement is felt (first stop). Grade III (high force) overcomes the “first stop” and is used to increase the mobility of periarticular tissues [29].

Imaging methods assess the motions that occur in the glenohumeral region [30,31]. Ultrasound is a real-time, non-invasive, and non-injurious imaging method used to assess patients compared with similar techniques [22,32]. The response variable to assess the effect of GLDM could be lateral movement [27,33,34]. This measurement has moderate to good reliability [35].

This study aimed to measure and compare the lateral motion of the humeral head and the rotational motion of the scapula when three different magnitudes of forces (low, medium, and high) were applied during GLDM, with and without scapular fixation.

## 2. Materials and Methods

### 2.1. Study Design

A cross-sectional study was conducted. A repeated measures design was used. The technique was applied to the GLDM in three force magnitudes (low, medium, and high) in two conditions (scapular fixation and non-fixation). The study variables were the lateral movement of the humeral head (coracohumeral distance) and the scapular rotation movement. The local ethical committee of Universitat Internacional de Catalunya approved this protocol (CBAS-2021-15). The study procedures were conducted following the Declaration of Helsinki (64th World Medical Association General Assembly Fortaleza, Brazil, October 2013). Informed consent was obtained from all participants.

### 2.2. Sample

Sample size calculation was carried out from the magnitude of the difference found in the variable lateral humeral head movement during high-force distraction mobilisations (d = 0.76) and the standard deviation (1.35 mm) of this variable based on a previous pilot study (*n* = 10 shoulders). The study had a significance level of 0.05, a power of 0.8, and no loss to follow up. Based on these parameters, the sample size consisted of 25 shoulders. The pilot study sample comprised five subjects (three men and two women). The procedure used in the pilot study was the same as the one used in the study.

The sample was recruited from volunteers at the Universitat Internacional de Catalunya. Inclusion criteria were as follows: subjects over 18 years of age who signed the informed consent form. Patients’ shoulders were excluded if they: (1) had pain in the shoulder region; (2) had a history of orthopaedic injuries in the shoulder region; (3) and were diagnosed with connective tissue involvement.

A total of 17 volunteers were recruited. Four volunteers were excluded because they had previously presented injuries to the shoulder (dislocations or fractures). In addition, one volunteer had a right shoulder with recent symptoms under study that made it inadvisable to perform the technique. The final sample consisted of 25 shoulders (13 left and 12 right) from 13 volunteers (7 men and 6 women) in fixation and non-fixation conditions. The mean age was 26.5 ± 9.3 years.

### 2.3. Experimental Procedure

The measurements were conducted in a single session. Socio-demographic characteristics were recorded at the beginning of the session: age, sex, arm dominance, height, weight, and body mass index.

All GLDM techniques were performed by a single physiotherapist with more than 20 years of clinical experience and knowledge of the applied technique.

A second physiotherapist, experienced in musculoskeletal ultrasound (more than five years), captured all ultrasound images. A third physiotherapist recorded the magnitude of force applied during low-, medium-, and high-force GLDM. He also measured the scapular rotational movement.

All subjects started with the scapular fixation condition. Scapular fixation was performed with the subjects in supine with a belt around the scapula and the thorax fixed in the opposite side of the treatment table. Two joint distraction belts were used. One was placed around the arm proximal to the shoulder of the patient. It was padded to avoid discomfort. The second one was placed around the pelvis of the physiotherapist performing the mobilisation (Figure 1A). Subjects were instructed to keep their arm relaxed. To measure the magnitude of force applied, a dynamometer (475055 Digital Force Gauge; Extech, Boston, MA, USA) was anchored between the two belts (Figure 1C).

The physiotherapist who applied the GLDM did not know the force (Newtons) exerted. He also did not have visual access to the ultrasound images. The third physiotherapist recorded the strength data and measured scapular rotation with a universal goniometer (Figure 1B). The glenohumeral joint was placed in its open-packed position (55° abduction and 30° horizontal arm adduction) [29]. A 40 mm linear transducer (USTTL01, 12L5) of a portable ultrasound machine (US Aloka Prosound C3 15.4) was placed transversely over the glenohumeral joint space (Figure 1D). The transducer was moved medially and laterally until the coracoid process and the superior aspect of the lesser tuberosity of the humerus were visible on the ultrasound image.

The mobilising therapist pulled the arm laterally [36], with three different magnitudes of GLDM force according to Kaltenborn grades of joint mobilisation movement with the glenohumeral joint in the open-packed position [29].

An ultrasound image and the scapular position were taken as baseline in the open-packed position of the shoulder without traction. Scapular rotational movement and the associated magnitude of force applied in the three magnitudes of GLDM were recorded. The low-force GLDM was determined when the physiotherapist verbally indicated that joint looseness had been picked up. The medium-force GLDM was when a marked resistance was first felt (the “first stop”), and the high-force GLDM was when there was maximum tissue resistance. This procedure was applied in the same sequence and was repeated twice with 30 s rest between repetitions [22]. The value of the applied force for the analysis was the average of the two tests.

After applying the GLDM technique in the scapular fixation condition, the GLDM without scapular fixation was performed in the same three force conditions. The subjects maintained the supine position with the glenohumeral joint in the open-packed position, but the axillary belt for scapular fixation was removed. A captured ultrasound image was taken, and the scapular position was taken as a baseline measurement in the non-scapular fixation condition. The magnitude of force applied during low, medium, and high GLDM in the non-scapular fixation condition were the mean values recorded during the scapular fixation condition. The mobilising therapist pulled until the third physiotherapist told him to stop because he had reached the desired strength. This force was the average force value used in the two repetitions of the fixation condition. The assessors recorded ultrasound images and scapular rotational movement twice at each GLDM force magnitude.

### 2.4. Ultrasound Measurements

A single examiner performed measurements of the echographic captures. This examiner was blind to what condition and force each image corresponded to. Ultrasound calibrated images were exported as jpg files and ImageJ (https://imagej.nih.gov/ij/docs/guide/ (accessed on 18 June 2023)) was used for all measurements. The ultrasound images were marked with a standard distance for subsequent measurement calibration. For each image, a line was drawn from the lateral aspect of the coracoid process to the medial aspect of the lesser tuberosity of the humerus [32,33,34]. This distance was measured in millimetres.

The amount of movement achieved with each mobilisation was calculated by subtracting the baseline resting distance from the distance achieved for each magnitude of force mobilisation (Figure 2). For the statistical analysis, we used the average distance recorded in the two trials.

Two assessments were performed on 10 subjects to determine the intra-observer reliability of the ultrasound image measurements. The subjects had the same characteristics as the study sample. The intraclass correlation coefficient (two-way mixed-effects model) (ICC_3,1_), standard error of measurement (SEM), and the minimum detectable change at the 95% confidence level (MDC95%) for ultrasound measurements of the lateral humeral head movement were calculated. All ultrasound measurements showed excellent reliability with an ICC_3,1_ value > 0.992 and an SEM of less than 0.04 mm. Full data are available in Appendix A.

### 2.5. Scapular Rotatory Movement Assessment

The baseline reference position was the open-packed position (55° abduction and 30° horizontal arm adduction) measured with a universal goniometer. The goniometer was placed in the centre of the glenohumeral joint on the dorsal side. The stationary goniometer arm was placed along the humerus diaphysis and the moving goniometer arm was aligned with the scapula’s lateral border. Changes in the different magnitudes of applied force were measured, following the same references. The variable’s mean value in the two trials for each scapular fixation condition and magnitude of applied force was used for statistical analysis.

### 2.6. Statistical Analysis

IBM SPSS Statistics for Windows, version 20.0 (Armonk, NY, USA: IBM Corp.), was used for all statistical analyses.

Descriptive statistics (mean and standard deviations, or number and percentage) were calculated to describe sample characteristics.

A linear mixed model (ANOVA) with a magnitude of force applied during GLDM (low, medium, and high) and scapular fixation condition (with fixation, without fixation) was used to analyse differences in lateral humeral head movement and scapular movement. If ANOVA was significant, Bonferroni-adjusted post hoc tests were used to assess pairwise comparisons. The same analysis was performed, segmented by sex and dominance. In addition, differences between sex and dominance for the same variables were analysed with the one-factor ANOVA test.

The effect size was calculated to estimate the magnitude of the difference between two conditions, in the main variables, with Cohen’s coefficient (d). Cohen’s coefficients were interpreted as follows: large effect sizes, d > 0.8; moderate effect sizes, d = 0.5–0.79; and small effect sizes, d = 0.2–0.49 [37].

## 3. Results

The demographic characteristics of the sample are shown in Table 1.

Table 2 provides the data for the magnitude of applied force, lateral humeral head movement, and scapular movement under scapular fixation and non-fixation conditions for each GLDM force magnitude, mean difference and 95% CI, effect sizes, and interaction effects. Two-way repeated-measures ANOVA showed a statistically significant interaction between the two factors on lateral humeral head movement (F = 28.850 *p* < 0.001) and scapular movement variable (F = 62.660, *p* < 0.001). Bonferroni post hoc tests revealed significant differences between scapular fixation and non-fixation conditions in lateral humeral head and scapular movement. Lateral humeral head movement was greater in all GLDM force applications in the scapular fixation condition compared with the non-scapular fixation condition. A statistically significant difference between the two conditions was reached in the medium-force (*p* = 0.010) and high-force (*p* < 0.001) applications. During the GLDM with high force, the most considerable difference was found between the scapular fixation and non-fixation condition. For scapular motion, values were remarkable for all GLDM force applications in the non-scapular fixation condition compared with the scapular fixation condition. The difference between the two fixation conditions was statistically significant in each force intensity application (*p* < 0.001). The angle remained more stable in the scapular fixation condition with 55° in the baseline until it reached 57.7 degrees. However, the non-scapular fixation condition changed from 55° degrees at baseline to 67.7° degrees at the high-force application.

The pattern was indistinctly related to sex and dominance. The most significant gain in the lateral motion of the humeral head was in the scapular fixation condition. For scapular rotation movement, it was the non-scapular fixation condition. There were no statistically significant differences between sexes in either variable. There was no statistically significant difference in lateral humeral head motion for comparison by dominance. Only the difference between scapular fixation conditions showed a statistically significant difference by dominance (*p* < 0.040) in scapular rotation. The largest difference was reached in the non-dominant shoulder. Full results, segmented by sex and dominance, are provided in Appendix A.

## 4. Discussion

To our knowledge, this is the first study to analyse the effect of scapular fixation on the lateral movement of the humeral head and the rotational movement of the scapula when applying three different magnitudes of forces during GLDM. The results of the present study showed that the lateral movement of the humeral head was significantly greater in the scapular fixation condition compared with the non-scapular fixation condition for all three magnitudes of GLDM force. Regarding rotational scapular movement, the opposite was found. A minimal movement was observed in the fixation condition. However, greater scapular rotational movement was observed in the non-scapular fixation condition.

The amount of lateral movement of the humeral head during GLDM was similar to that found in the study by Guerra-Rodríguez et al. [32]. They found a movement gain of 3.97 ± 0.24 mm in a high-force application in a non-joint fixation condition. At the same time, we achieved a distance of 2.34 ± 1.77 mm in the same condition. The difference is that they did not quantify the force applied, so we cannot be sure that their high force was of greater magnitude. However, the results are more similar to what we found in our study in the joint fixation condition, where the values we achieved in a high-force magnitude were 3.72 ± 1.95 mm. The position differs slightly as their resting position is at 25° of abduction whereas ours is 55° of shoulder abduction [29]. In addition, interestingly, they obtained a greater distance at the “Zero position” (arm along the body) (5.74 ± 0.51 mm), where capsular structures should not be at their maximum relaxation. We decided to keep the position of 55° abduction as this is as described by Kaltenborn [29].

During the high-force GLDM application, both scapular fixation conditions showed an increase in scapular rotatory movement. However, it was much less in the fixation condition, with a 2.70 ± 2.37° difference compared with 12.71 ± 6.12° in the non-fixation condition. These results are similar to those found in other glenohumeral axial distraction studies [22,27]. Garwood et al. have already described scapular rotational movement during the application of distraction techniques without fixation [38]. No statistically significant differences were found between the same variables by sex. However, a statistically significant difference in dominance was found in scapular rotation. It was found that the non-dominant shoulder showed a more significant difference in scapular rotation at high-force application. This greater difference may be due to less tone in the non-dominant shoulder, which causes the non-dominant shoulder to drag more on the shoulder without scapular fixation.

Following Kaltenborn’s indications [29], traction techniques using low and medium forces would be indicated for pain relief. However, the differences in joint rotation and distraction at these force intensities are slight between the two fixation conditions. According to Kaltenborn [29], the high-force application is indicated for tissue elongation. This application achieves greater joint distraction and tissue elongation [12,17]. Therefore, it is essential to avoid rotational movement of the scapula, which would cause a change in the perpendicularity of the technique. Correct scapular fixation would focus the force on the desired tissue and avoid possible adverse effects due to force transmission to unwanted regions [23,24,25]. In addition, the scapular fixation condition is the one that achieves the greatest coracohumeral distraction with the same force application. These results are similar between sexes, with slight variations in the magnitudes of force applied.

In this study, we did not assess the structure(s) receiving the stress during GLDM application. We saw a separation of the lateral movement, but we did not know which structure stopped the movement. Studies similar to those performed by Estébanez-de-Miguel et al. [39,40,41] in the hip region would need to know the amount of stress received by the different structures and its relationship to the force applied during mobilisation. We would also need to know what effect is produced in lateral movement in positions other than the resting position described by Kaltenborn [29], where these treatment techniques are applied. Theoretically, traction out of the resting position would involve greater ligament tension and, thus, less movement. However, in the study by Guerra-Rodríguez et al. [32], they found greater movement in the zero position than in the resting position.

In a recent study by Estebánez-de-Miguel et al. [42], they found that, to achieve tissue elongation in the hip joint, at least 45 s were needed. No data have been found on the glenohumeral joint, but the time may be similar. Therefore, an external aid such as a traction strap may be necessary to maintain strength. Furthermore, with the data obtained in this study, we can affirm that scapular fixation achieves greater control of scapular position and joint distraction and can be maintained over time with the help of the traction strap.

The GLDM force magnitudes used in the present study were similar to the study by López-de-Celis et al. [22], with axial distraction, and higher than the study by Witt et al., 2016 [27], although the latter applied caudal distraction. In the study by López-de-Celis et al. [22] the force intensity in low and medium force was lower than those in the present study, with values of 16.2 N ± 5.1 for grade I and 46.7 N ± 12.4 for grade II, whereas in the present study the applied force values were 22.4 N ± 3.8 and 68.0 N ± 9.6, respectively. However, for high force, a higher force was applied in the present study, with a difference of 57.8 N. We cannot compare this with the applied force in the study by Guerra-Rodríguez et al. [32], who, with practically equal coracohumeral distraction distances, did not evaluate the applied force.

Although the subjects were instructed to remain relaxed, we cannot rule out that some defensive contractions may have occurred during the application of the technique. It has been described that there is a reactive contraction in the contractile tissues as a reactive force to any load applied to the glenohumeral joint [35]. This defensive contraction may have minimised the sideways movement, so the results may have been lower than expected. However, we consider the conditions applied to be more similar to standard clinical practice. Other studies have already evaluated the reliability of detecting degrees of movement, showing good to excellent intra-observer reliability [39,43,44,45,46,47]. Ultrasound measurements are reliable for assessing inferior gliding of the humeral head [22,27,28,35]. The present study observed excellent intra-observer reliability (ICC greater than 0.992), exceeding SEM in all applied force magnitudes.

This study has limitations. Firstly, for methodological reasons, there was no randomisation of the techniques to scapular or non-scapular fixation conditions. Therefore, the effect of repetition on the shoulder tissues could have influenced the measurements, and the differences found could have been more significant. Secondly, since the study was performed on asymptomatic subjects, the results obtained in the present study cannot be extrapolated to patients with pathology. Thirdly, despite informing the subjects that they should leave the arm relaxed during the application of the GLDM, we cannot be sure that there was not some defence contraction that may have minimised the results. Finally, a reliability of the measurement of scapular rotational movement was not performed. Therefore, the exact results may vary slightly from those presented in the study.

## 5. Conclusions

Scapular fixation resulted in more significant lateral movement of the humeral head than in the non-scapular fixation condition during three intensities of GLDM forces. The scapular position did not change during GLDM with the scapular fixation condition.

## Figures and Tables

**Figure 1 healthcare-11-01829-f001:**
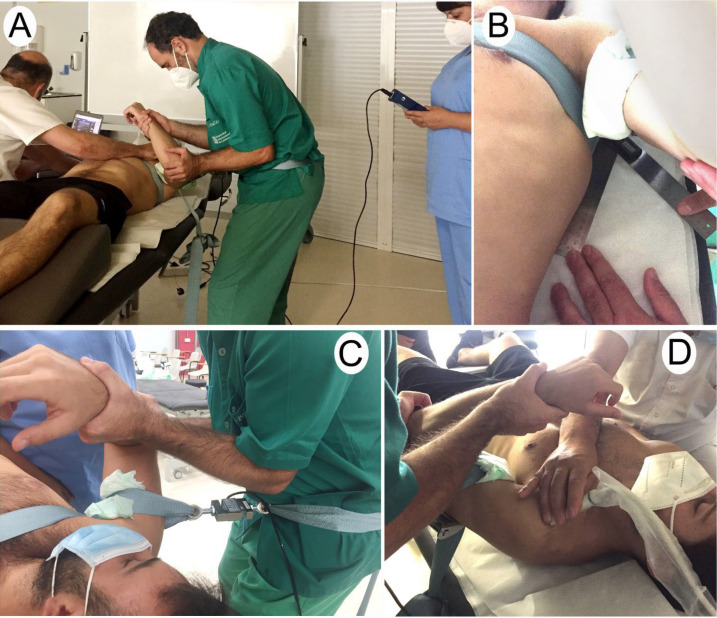
(**A**) Experimental set-up: glenohumeral lateral traction mobilisation technique with scapular fixation condition; (**B**) Placement of the goniometer; (**C**) Dynamometer attached to the belt; (**D**) Transducer placed transversely over the glenohumeral joint.

**Figure 2 healthcare-11-01829-f002:**
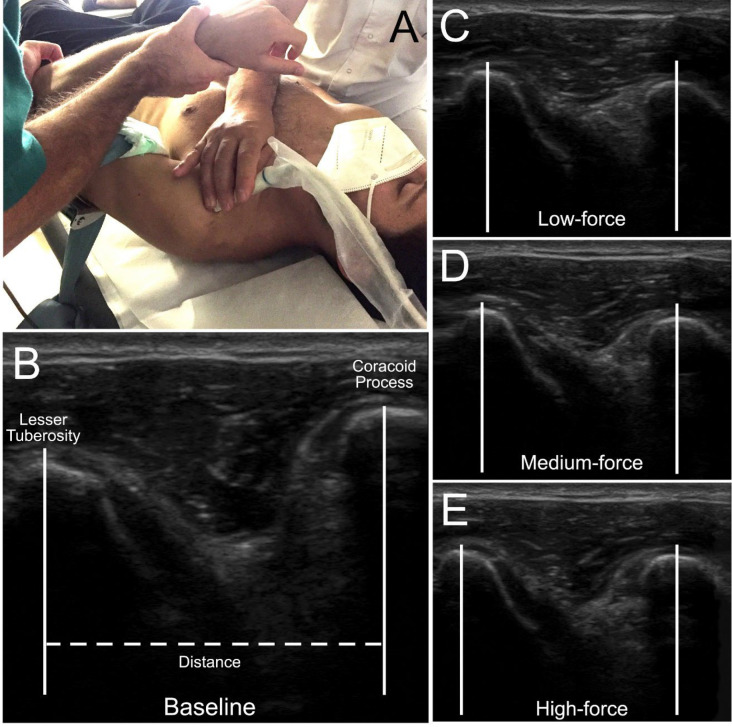
(**A**) Transducer position; (**B**) ultrasound image at rest (baseline) with the landmarks of the coracohumeral process and lesser tuberosity identified; (**C**) ultrasound image of humeral head position in low-force GLDM in scapular fixation; (**D**) ultrasound image of the position of the humeral head in GLDM of medium force in scapular fixation; (**E**) ultrasound image of the humeral head position in GLDM of high force in scapular fixation.

**Table 1 healthcare-11-01829-t001:** Subjects’ demographic characteristics.

	Mean ± SD or *n* (%)
Age (year)	26.5 ± 9.3
Gender	
Men	7 (53.8%)
Women	6 (46.2%)
Dominance	
Right	11 (84.6%)
Left	2 (15.4%)
Height (cm)	169.1 ± 10.3
Weight (kg)	60.7 ± 21.6
BMI (kg/m^2^)	24.0 ± 5.2

Abbreviations: SD, standard deviation; cm, centimetre; kg, kilogram; BMI, body mass index; *n*, number.

**Table 2 healthcare-11-01829-t002:** Outcomes of the magnitude of force applied with GLDM, with and without scapular fixation, in lateral movement of the humeral head and scapular movement.

Variable	Magnitude of GLDM Force	Scapular Fixation	Non-Scapular Fixation	Mean Difference (95%CI)	Effect Size	*p*-Value
Lateral movement of the humeral head	Low-force (22.4 ± 3.8 N)	1.00 ± 0.65 mm	0.79 ± 0.79 mm	0.22 mm (−0.13, 0.56) *p* = 0.212	0.29	F = 28.850 *p* < 0.001
	Medium-force (68.0 ± 9.6 N)	2.17 ± 1.14 mm	1.64 ± 1.32 mm	0.54 mm (0.14, 0.93) *p* = 0.010	0.43	
	High-force (143.2 ± 20.0 N)	3.72 ± 1.95 mm	2.34 ± 1.77 mm	1.38 mm (0.94, 1.83) *p* < 0.001	0.74	
Scapular movement	Low-force (22.4 ± 3.8 N)	55.5 ± 0.8°	56.7 ± 1.7°	−1.3° (−1.91, −0.63) *p* < 0.001	0.97	F = 62.660 *p* < 0.001
	Medium-force (68.0 ± 9.6 N)	56.3 ± 1.3°	60.8 ± 3.4°	−4.5° (−5.72, −3.28) *p* < 0.001	1.73	
	High-force (143.2 ± 20.0 N)	57.7 ± 2.4°	67.7 ± 6.1°	−10.0° (−12.25, −7.78) *p* < 0.001	2.16	

GLDM, glenohumeral lateral distraction mobilisation; N, Newtons; mm, millimetres; °, grades.

## Data Availability

The data presented in this study are available on request from the corresponding author.

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
