# Peer review of "Effect of Scapular Fixation on Lateral Movement and Scapular Rotation during Glenohumeral Lateral Distraction Mobilisation"

_healthcare, 2023, doi:10.3390/healthcare11131829_

Round 1
Reviewer 1 Report
First of all, thank you for your fascinating paper about an infrequent topic. It was interesting for me, but I have a few suggestions, which you will find below.
Abstract:
Line 18: „Upper limbs were selected” – on the pictures, it seems the authors used participants and not just limbs.
Introduction:
It is quite confusing; the author writes about the incidence of shoulder injuries in general and then mentions adhesive capsulitis. It is unclear if the distraction technique is meant to be for all shoulder pathologies, especially adhesive capsulitis. If so, the percentage for adhesive capsulitis would be more interesting.
Line: 53 – interpunction is not correct. Please do proofreading before submitting.
Please explain briefly, as some readers might need to become more familiar with the Kaltenborn concept.
Material and Methods
Line 69:Please perform proofreading more carefully.
Please define the inclusion criteria in more detail, as participants with shoulder pain and a history of shoulder injuries were included, but four patients were excluded because of injuries.
The force levels are described as very vague. Are there any measurement ranges for low to high according to the dynamometer? Otherwise, it is not reproducible for other study groups. Why was it necessary to blind the physiotherapist for the dynamometer? A second investigator would be necessary, at least. In the discussion section, you mention values for the three levels. Why did you not choose those values for your groups?
Line 163: Please proofread before submitting
Line 172: Trnasducer - please proofread before submitting
Results:
Were there any differences between the dominant and non-dominant shoulders?
Discussion:
It is well-written but needs some points. The clinical usage should be highlighted, and the few other papers should be discussed and compared in more detail.
General:
A second investigator would be beneficial and reduce bias. Side differences (dominant shoulder) would be of great interest. As there are only 13 participants, their data could be presented in a separate table.
All in all, this paper has many methodological flaws; therefore, intensive revision is needed.
English proof-reading is necessary.
Please do spell- and grammar checks before submitting
Author Response
We thank the reviewer for his time and dedication to improving this manuscript.
Review 1
First of all, thank you for your fascinating paper about an infrequent topic. It was interesting for me, but I have a few suggestions, which you will find below.
Thank you for your comment.
Abstract:
Line 18: „Upper limbs were selected” – on the pictures, it seems the authors used participants and not just limbs.
Thank you. This sentence has been modified in the abstract.
“Seventeen volunteers were recruited (n=25 upper limbs)”
Introduction:
It is quite confusing; the author writes about the incidence of shoulder injuries in general and then mentions adhesive capsulitis. It is unclear if the distraction technique is meant to be for all shoulder pathologies, especially adhesive capsulitis. If so, the percentage for adhesive capsulitis would be more interesting.
Thank you for your observation.
The proposed technique is generally indicated in mobility restrictions. The therapist graduates the force according to the target. One of the pathologies where the technique is indicated is adhesive capsulitis. We have added in the introduction information on the prevalence of adhesive capsulitis. In addition, we have also added in the introduction that it would be indicated in shoulder mobility dysfunctions.
“Mobility dysfunctions in the shoulder, such as adhesive capsulitis, are estimated to be prevalent in 2% to 5% of the general population.”
- D'Orsi GM, Via AG, Frizziero A, Oliva F. Treatment of adhesive capsulitis: a review. Muscles Ligaments Tendons J. 2012 Sep 10;2(2):70-8. PMID: 23738277; PMCID: PMC3666515.
Line: 53 – interpunction is not correct. Please do proofreading before submitting.
Thank you for your input. The entire manuscript has been revised.
Please explain briefly, as some readers might need to become more familiar with the Kaltenborn concept.
Thanks, you for your observation. Thanks to your suggestion, we have added the information of Kalternbon concept in the text.
“According to Kaltenborn, the traction force grading system consists of applying a manual force on the joint perpendicular to the concave surface. Grade 1 corresponds to a small tensile force that cancels out the natural compressive forces of a joint. Grade 2 is a force that tightens the periarticular tissue to the so-called "first stop", stretching the tissues to the limit and feeling the maximum resistance to movement. Grade 3 is a force that exceeds the "first stop" and is used to increase the mobility of the periarticular tissues.”
Material and Methods
Line 69:Please perform proofreading more carefully.
Thank you. Errors have been corrected.
Please define the inclusion criteria in more detail, as participants with shoulder pain and a history of shoulder injuries were included, but four patients were excluded because of injuries.
Thank you. The exclusion criteria have been improved as suggested by the reviewer.
“Upper limbs were excluded if they had: 1) Pain in the shoulder region; 2) History of orthopaedic injuries in the shoulder region; 3) And volunteers diagnosed with connective tissue involvement.”
The force levels are described as very vague. Are there any measurement ranges for low to high according to the dynamometer? Otherwise, it is not reproducible for other study groups.
We have added the Kaltenborn force application system explanation so that readers can better understand the graduation. In other articles, "low, medium, and high" are used, so we have kept this terminology, which we consider universal.
Why was it necessary to blind the physiotherapist for the dynamometer?
Thank you for your comment.
During clinical practice the force applied is based on the sensation perceived by the therapist, so we believe it is necessary to blind the physiotherapist to the force applied keeping it like in clinical practice. This way has proven reliable in detecting the force in the three magnitudes
- Estébanez-de-Miguel E, López-de-Celis C, Caudevilla-Polo S, González-Rueda V, Bueno-Gracia E, Pérez-Bellmunt A. The effect of high, medium and low mobilization forces applied during a hip long-axis distraction mobilization on the strain on the inferior ilio-femoral ligament and psoas muscle: A cadaveric study. Musculoskelet Sci Pract. 2020 Jun;47:102148.
A second investigator would be necessary, at least.
Exactly. We have already explained that a third Physiotherapist (Fig 1 B) collected the applied force data and then measured the scapular rotation with the goniometer.
In the discussion section, you mention values for the three levels. Why did you not choose those values for your groups?
Thank you for your comment. We do not use the values from the Guerra-Rodriguez et al. article as they perform manual traction, and we have performed it with distraction belts. The belt technique is how it is applied in the clinic. It allows us to maintain adequate traction time in an actual patient.
Line 163: Please proofread before submitting
Thank you for your comment.
The full text of the manuscript has been revised.
Line 172: Trnasducer - please proofread before submitting
The typographical mistake has been corrected.
Results:
Were there any differences between the dominant and non-dominant shoulders?
Thank you for your comment.
Statistical analysis has been conducted to assess whether there are differences between limbs according to dominance and sex.
The pattern is similar to the global values, both by dominance and sex. In the comparison between dominant and non-dominant limbs, there is only a statistically significant difference in the high traction force in the scapular movement.
We will include the tables as supplementary material and have added information on this new analysis in the text.
Discussion:
It is well-written but needs some points. The clinical usage should be highlighted, and the few other papers should be discussed and compared in more detail.
Thank you for your comments.
Changes have been made throughout the discussion section, on clinical aspects and in comparison with other studies.
General:
A second investigator would be beneficial and reduce bias.
Thank you for your comments.
We understand this to refer to a single researcher who applied the technique under study.
The reliability of detection of the grades of movement has already been evaluated in other studies, showing a good or an excellent intra-observer.
1.- Estébanez-De-Miguel, E.; López-De-Celis, C.; Caudevilla-Polo, S.; González-Rueda, V.; Bueno-Gracia, E.; Pérez-Bellmunt, A. The effect of high, medium and low mobilization forces applied during a hip long-axis distraction mobilization on the strain on the inferior ilio-femoral ligament and psoas muscle: A cadaveric study. Musculoskelet. Sci. Pract. 2020, 47, 102148.
2.- Estébanez-De-Miguel, E.; Fortún-Agud, M.; Jimenez-Del-Barrio, S.; Caudevilla-Polo, S.; Bueno-Gracia, E.; Tricás-Moreno, J.M. Comparison of high, medium and low mobilization forces for increasing range of motion in patients with hip osteoarthritis: A randomized controlled trial. Musculoskelet. Sci. Pract. 2018, 36, 81–86.
3.- Vermeulen, H.M.; Rozing, P.M.; Obermann, W.R.; le Cessie, S.; Vlieland, T.V. Comparison of High-Grade and Low-Grade Mobilization Techniques in the Management of Adhesive Capsulitis of the Shoulder: Randomized Controlled Trial. Phys. Ther. 2006, 86, 355–368.
4.- Maher, S.; Creighton, D.; Kondratek, M.; Krauss, J.; Qu, X. The effect of tibio-femoral traction mobilization on passive knee flexion motion impairment and pain: A case series. J. Man. Manip. Ther. 2010, 18, 29–36.
5.- Courtney, C.A.; Steffen, A.D.; Fernández-De-Las-Peñas, C.; Kim, J.; Chmell, S.J. Joint Mobilization Enhances Mechanisms of Conditioned Pain Modulation in Individuals with Osteoarthritis of the Knee. J. Orthop. Sports Phys. Ther. 2016, 46, 168–176.
6.- Courtney, C.A.; Witte, P.O.; Chmell, S.J.; Hornby, T.G. Heightened Flexor Withdrawal Response in Individuals with Knee Osteoarthritis Is Modulated by Joint Compression and Joint Mobilization. J. Pain 2010, 11, 179–185.
7.- Park, S.-S.; Kim, B.-K.; Moon, O.-K.; Choi, W.-S. Effects of joint position on the distraction distance during grade III glenohumeral joint distraction in healthy individuals. J. Phys. Ther. Sci. 2015, 27, 3279–3281.
Side differences (dominant shoulder) would be of great interest. As there are only 13 participants, their data could be presented in a separate table.
Thank you for your comment.
We have added this information in the text and tables in the supplementary material.
All in all, this paper has many methodological flaws; therefore, intensive revision is needed.
Thank you for your comments. We hope that the modifications made to the manuscript have improved the manuscript sufficiently for publication.
English proof-reading is necessary.
Please do spell- and grammar checks before submitting
The article has been spell-checked and grammatically checked by a native speaker from the University—apologies for any initial errors.
Reviewer 2 Report
In general, I appreciate this interesting approach: a technique that is widely applied but never measured/quantified, so let's measure the effects of GLDM. However, the manuscript could be written in a more scientific fashion:
Over the several sections, there is quite some repetition, limiting readability.
When numbers are mentioned, please add the units of measurements.
Several topics are mixed throughout the several sections of the methods, please re arrange.
Too many sentences throughout the manuscript are just too long to be read without taking notes, and regularly contain multiple messages. Please adapt into a readable and concise style.
page2_line52: "we believe that…" A scientific publication is not about believes, but facts and evidence.
page2_line86-69: need rephrasing, as it contains several repetitions, without making completely clear what design is applied.
page4_line152: from the photos it is clearly visible what the fixation condition was, how could you possibly "blind" the UltraSound examiner for such. Or was the one taking the ultrasound during the experiment not the examiner of the images? Then you should describe that.
Oh okay, now I see it described in experimental procedures, but then why have a separate section for ultrasound measurements?
page5_line: Table 1 shows results and should therefore be presented in the results section. Or, since it addresses intra-observer reliability, consider adding this as an Appendix / supplement.
page6_line181: placing a universal goniometer in the centre of the glenohumeral joint at the dorsal side of a participant lying on its back is merely based on guessing & gambling then measuring. At this point I really doubt the appropriateness of the method applied, and if the procedure can be reliable at all.
page7_line213: Either this line has an error or this finding is in contradiction with your conclusion. Furthermore, you can't apply all three magnitudes of GLDM forces at the same time, so what exactly do you mean here.
page7_line222: with the described methods the presented accuracy of two digits for angles is a fake accuracy. (Yes, over a given number of participants you might get a higher accuracy due to averaging, but two digits is not realistic)
page8_line281: This would have minimised lateral movement so that the results could be more significant. In what way? I would expect the opposite, protective/reactive contraction of contractile tissue, less displacement….
It is not in the grammar but merely the style of writing, with a lot of repetitions throughout the manuscript. Also the mix of topics throughout the sections should be addressed.
Author Response
We thank the reviewer for his time and dedication to improving this manuscript.
Review 2
In general, I appreciate this interesting approach: a technique that is widely applied but never measured/quantified, so let's measure the effects of GLDM. However, the manuscript could be written in a more scientific fashion:
Thank you for your comments. We look forward to improving the article for publication.
Over the several sections, there is quite some repetition, limiting readability.
All changes requested by the reviewers have been made. In addition, to improve readability, information duplicity has been checked.
When numbers are mentioned, please add the units of measurements.
The manuscript has been revised, and units of measurement have been added where missing.
Several topics are mixed throughout the several sections of the methods, please re arrange.
Thank you for your feedback. We have revised the manuscript.
Too many sentences throughout the manuscript are just too long to be read without taking notes, and regularly contain multiple messages. Please adapt into a readable and concise style.
Thank you for your comments.
The entire manuscript has been revised to improve readability.
page2_line52: "we believe that…" A scientific publication is not about believes, but facts and evidence.
Thank you for your review. We have modified this expression.
page2_line86-69: need rephrasing, as it contains several repetitions, without making completely clear what design is applied.
Your comment is welcome. We have corrected the error and revised the wording.
page4_line152: from the photos it is clearly visible what the fixation condition was, how could you possibly "blind" the UltraSound examiner for such. Or was the one taking the ultrasound during the experiment not the examiner of the images? Then you should describe that.
Thank you for your comment.
We have clarified this situation in the manuscript. The person performing the ultrasound scan took the image and made a screenshot. The images were coded. The measurements of the coded images were made later by an evaluator who did not know which procedure they belonged to.
Oh okay, now I see it described in experimental procedures, but then why have a separate section for ultrasound measurements?
Thank you for your comment.
We have made a separate section to evaluate the ultrasound images from the general procedure. It allows us to explain better how the measurement of the ultrasound image captures was performed and its reliability. We have revised the wording to make it easier to read.
page5_line: Table 1 shows results and should therefore be presented in the results section. Or, since it addresses intra-observer reliability, consider adding this as an Appendix / supplement.
Thank you for your input.
Your suggestion seems appropriate. We have added a general comment on the reliability of the measurements in the text of the manuscript. Table 1 has been removed and will be moved to supplementary material.
page6_line181: placing a universal goniometer in the centre of the glenohumeral joint at the dorsal side of a participant lying on its back is merely based on guessing & gambling then measuring. At this point I really doubt the appropriateness of the method applied, and if the procedure can be reliable at all.
Thank you for your comment.
Indeed, the goniometric measurement in this position could be better. We did not have any other measurement method that respects the procedure as it is performed in the clinic. We have added this aspect to the limitations.
"Finally, a reliability of the measurement of scapular rotational movement was not performed. Therefore, the exact results may vary slightly from those presented in the study".
page7_line213: Either this line has an error or this finding is in contradiction with your conclusion. Furthermore, you can't apply all three magnitudes of GLDM forces at the same time, so what exactly do you mean here.
Thanks to the reviewer for making us aware of the drafting error.
It is not possible to apply all three force magnitudes at the same time. The wording has been modified.
page7_line222: with the described methods the presented accuracy of two digits for angles is a fake accuracy. (Yes, over a given number of participants you might get a higher accuracy due to averaging, but two digits is not realistic)
Thank you for your comment.
The table has been modified by reducing the scapular rotation movement values to one digit. This format has been respected in the supplementary sex and dominance segmentation tables suggested by reviewer 1.
page8_line281: This would have minimised lateral movement so that the results could be more significant. In what way? I would expect the opposite, protective/reactive contraction of contractile tissue, less displacement….
Thanks again for making us aware of the misinterpretation. Precisely what the reviewer comments is what we wanted to have expressed. We have modified the wording to clarify this.
“This could have minimized lateral movement, so the results may have been lower than expected.”
it is not in the grammar but merely the style of writing, with a lot of repetitions throughout the manuscript. Also the mix of topics throughout the sections should be addressed.
Thank you for your comment.
We have thoroughly revised the wording of the manuscript. The article has been spelled and grammatically checked by a native speaker from the University. We apologise for any initial errors.
Round 2
Reviewer 1 Report
The authors have adressed my concerns.
Still minor grammar issues
Author Response
Thank you for your comments and help in improving the manuscript.
Reviewer 2 Report
The authors have thoroughly adapted the manuscript based on the reviewers comments, compliments. Readability is certainly improved, and procedures and limitations thereof better explained.
Only two minor comments left from my side:
In alignment with the comment of the other reviewer, at several locations in the manuscript the phrasing "limbs" is still used, while at other paragraphs "shoulder" is used. I would suggest to consistently use "shoulder", or "patients' shoulder" or "shoulder of the patient" as it is the focus of the paper.
line 163-166: this assumes the US images taken are calibrated. This means that either a reference marker with known measures is in the US image, or the system itself delivers calibrated images, which to my knowledge, is not the case in US. It is a minor technical detail but becomes relevant when quantifying measures from US. Although the ICC and SEM are indeed wonderful :-)
Please address this in the limitation section. See Data collection: Ultrasound examinations from [1] as an example.
1. Bossuyt FM, Boninger ML, Cools A, Hogaboom N, Eriks-Hoogland I, Arnet U, et al. Changes in supraspinatus and biceps tendon thickness: influence of fatiguing propulsion in wheelchair users with spinal cord injury. Spinal Cord. 2020;58(3):324-33.
But if addressed, I'm fine and will recommend acceptance of the manuscript!
Author Response
Review 2
The authors have thoroughly adapted the manuscript based on the reviewers comments, compliments. Readability is certainly improved, and procedures and limitations thereof better explained.
Thank you for your comments and help in improving the manuscript.
Only two minor comments left from my side:
In alignment with the comment of the other reviewer, at several locations in the manuscript the phrasing "limbs" is still used, while at other paragraphs "shoulder" is used. I would suggest to consistently use "shoulder", or "patients' shoulder" or "shoulder of the patient" as it is the focus of the paper.
Thank you. Expressions referring to limbs have been modified and replaced by shoulder
line 163-166: this assumes the US images taken are calibrated. This means that either a reference marker with known measures is in the US image, or the system itself delivers calibrated images, which to my knowledge, is not the case in US. It is a minor technical detail but becomes relevant when quantifying measures from US. Although the ICC and SEM are indeed wonderful :-)
Please address this in the limitation section. See Data collection: Ultrasound examinations from [1] as an example.
- Bossuyt FM, Boninger ML, Cools A, Hogaboom N, Eriks-Hoogland I, Arnet U, et al. Changes in supraspinatus and biceps tendon thickness: influence of fatiguing propulsion in wheelchair users with spinal cord injury. Spinal Cord. 2020;58(3):324-33.
Thank you for the reviewer's comment.
This information needed to be clarified. Information has been added in the manuscript in the "Ultrasound measurements" section about the calibration of the ultrasound image.
“The ultrasound images were marked with a standard distance for subsequent measurement calibration.”
But if addressed, I'm fine and will recommend acceptance of the manuscript!